

# Mediating roles of character traits and parenting in the relationship between maternal effortful control and children's conduct problems

Maor Yeshua[1], Ada H. Zohar[2,3,4] and Andrea Berger[1,5]

[1] Department of Psychology, Ben-Gurion University of the Negev, Beer Sheva, Israel
[2] Department of Behavioral Sciences, Ruppin Academic Center, Emek Hefer, Israel
[3] Graduate Program in Clinical Psychology, Ruppin Academic Center, Emek Hefer, Israel
[4] Graduate Program in Gerontological Clinical Psychology, Ruppin Academic Center, Emek Hefer, Israel
[5] School of Brain Sciences and Cognition, Ben-Gurion University of the Negev, Beer Sheva, Israel

Corresponding author
Maor Yeshua,
Maor.yeshua@gmail.com

## ABSTRACT

**Background:** Parenting practices are crucial to children's development and are important predictors of children's conduct problems. The aim of the current study was to test the mediating role of mothers' character traits on the relationship between their temperamental self-regulation and their parenting practices, and on their children's conduct problems.

**Method:** A representative sample of 387 Israeli mothers of kindergarten children was recruited online. They completed questionnaires about their own effortful control (adult temperament questionnaire; ATQ), character traits (temperament and character inventory-revised (TCI-R), big five inventory (BFI)), and parenting practices (coping with children's negative emotions scale; CCNES), as well as conduct problems of their children (strengths and difficulties questionnaire; SDQ). Structural equation models were fitted, testing for direct and indirect connections, once with character traits drawn from the TCI and once with BFI traits.

**Results:** In both analyses, the first model presented a significant direct effect between mothers' effortful control and children's conduct problems. When including mother's parenting and character (based on the TCI or on the BFI) in the model, the direct path became insignificant and significant mediation effects were found; specifically, the indirect path through the parenting practices, as well as the mediated mediation path through the parenting practices and character. Moreover, mediation effects were found between mothers' effortful control and parenting practices through some character traits. The selected models showed a good fit (*e.g.*, NFI = 0.985; CFI = 0.997; RMSEA = 0.038).

**Discussion:** Our findings emphasize the importance of the mother's mature personality characteristics, the mother's actual parental practices, and the crucial value of this path for predicting child behavior outcomes.

## INTRODUCTION

An extensive theoretical and empirical literature supports the significance of parenting practices on the development of children's self-regulation and behavior (see reviews by *Bridgett et al. (2015)*, *Cabrera, Volling & Barr (2018)*, *Morawska (2020)*). The literature usually addresses parenting practices in terms of particular practices (such as warmth, hostility, laxness, *etc.*; *Bornstein, Hahn & Haynes, 2011*; *de Haan, Deković & Prinzie, 2012*; *Kitamura et al., 2009*; *Shaffer et al., 2018*), or as parental styles (such as authoritarian, authoritative or permissive; *Huver et al., 2010*; *Jones et al., 2014*; *Zohar, Lev Ari & Bachner-Melman, 2018*). However, it is possible to collapse all these terminologies into two main categories: adaptive and maladaptive parenting practices, depending on their implication for the child.

One of the foundations of parenting practices is the parents' self-regulation (*Baker & Brooks-Gunn, 2020*; *Bridgett et al., 2015*). Self-regulation refers to the flexible regulation of cognition, action, and emotion (*Berger, 2011*; *Nigg, 2017*; *Posner & Rothbart, 2000*), including conscious or unconscious efforts to control one's inner state (*Vohs & Baumeister, 2016*). Self-regulation develops and consolidates mainly during childhood as neurocognitive and neuro-emotional systems develop and interact with the environment, resulting in a positive or negative developmental cascade (*Nigg, 2017*; *Posner & Rothbart, 2000*; *Sapienza & Masten, 2011*). Temperamental self-regulation, generally referred to as effortful control (EC), is mainly composed of inhibitory control, attention shifting, and perceptual sensitivity (*Eisenberg, 2017*; *Rothbart & Bates, 2006*; *Rothbart et al., 2003*). Kindergarten age is a critical point in the development of EC (*Berger, 2011*) and individual differences at this age predict academic success, externalizing problems and other important aspects of life (*Eisenberg et al., 2005*; *Lee, Chang & Olson, 2022*; *Liew et al., 2008*; *Zhou, Main & Wang, 2010*). Hence, the relationship between parents and their kindergarten-age children is the focus of the current study.

### Effortful control and parenting practices

Parents' ability to control their thoughts, emotions, and behaviors is important for their ability to exert appropriate parenting practices (*Rutherford et al., 2015*; *Sanders & Mazzucchelli, 2013*; *Sanders & Morawska, 2018*; *Sanders, Turner & Metzler, 2019*). Well-regulated parents are likely to experience more confidence in their parenting ability and are more likely to be supportive and sensitive to their child's needs (*Distefano et al., 2018*; *Sanders, Turner & Metzler, 2019*; *Shaffer & Obradović, 2017*). Their parenting is more sensitive and involved (*Crandall, Deater-Deckard & Riley, 2015*; *Lorber, 2012*), yet they clearly establish their expectations for their children and encourage them to self-regulate themselves (*Shaffer et al., 2018*).

In contrast, research has demonstrated that a low level of parental self-regulation is related to an indulgent parenting style (*Crandall et al., 2016*) as well as to difficulties in parents mentalizing their child's emotional state and adjusting their behavioral response to their child's affective needs (*Jones et al., 2014*; *Schultheis, Mayes & Rutherford, 2019*). Alongside these findings, poor parental EC has been indicative of frequent engagement in
dysphoric, lax, over-reactive, and ineffective parenting (*Bridgett et al., 2011*, *2013*; *Davenport et al., 2011*).

Yet, parental practices are not solely determined by parental self-regulation. Additional and important aspects of human personality may be involved in parenting and can mediate the relationship between EC and parenting practices. For example, acceptance of others, which is an important element of cooperativeness (*Cloninger, Svrakic & Przybeck, 1993*; *Cloninger & Svrakic, 1997*), seems to be highly relevant for parenting practices. However, and quite surprisingly, as far as we know, no research has been conducted so far that explores the relation among the three variables of cooperativeness, self-regulation, and parenting.

## Character traits

Among the character traits that might be relevant for the theoretical framework linking EC and parenting practices, the current study will focus on two traits that are the mark of mature personality and are central to the temperament and character model (*Cloninger, Svrakic & Przybeck, 1993*; *Cloninger & Svrakic, 1997*) and included in the big five model (*Soto & John, 2017*). Cooperativeness (TCI-R) or agreeableness (BFI) is the willingness to help others, being accepting of others, compassionate, and generally socially tolerant (*Cloninger, Svrakic & Przybeck, 1993*; *Cloninger & Svrakic, 1997*; *Soto & John, 2017*; *Zohar & Cloninger, 2011*); while self-directedness is being responsible, purposeful, resourceful, self-accepting, disciplined, and organized, as measured by the TCI and the TCI-R (*Cloninger, Svrakic & Przybeck, 1993*; *Cloninger & Svrakic, 1997*; *Zohar & Cloninger, 2011*) and by conscientiousness in the big-five (*Soto & John, 2017*).

The use of both the TCI-R and the BFI in the current study is motivated by the fact that although there is overlap between the traits described, there are also differences (*Capanna et al., 2012*). Conscientiousness focuses on being hardworking, persistent and reliable (*i.e.*, having good working order and discipline) and less on the self-determination of the goals driving the individual's behavior (these behavioral emphases are better captured by the TCI temperamental trait of persistence; *Cloninger et al., 2012*). Self-directedness also includes self-acceptance, initiative and resourcefulness that are not included in conscientiousness. Agreeableness includes behaving in pleasant and non-aggressive ways that support working with others harmoniously, while cooperativeness describes accepting and helping others and acting toward them in a principled way (*Cloninger & Svrakic, 1997*; *Soto & John, 2017*).

## Effortful control and cooperativeness/agreeableness

High cooperativeness is defined as being socially tolerant and tenderhearted, empathic, helpful, and compassionate; desiring to help in the presence of unconditional acceptance of others, and willingness to help without a desire for selfish domination (*Cloninger & Svrakic, 1997*). *Evans & Rothbart (2007)*, as well as *Cloninger et al. (2019)*, did not find a significant correlation between EC and cooperativeness. However, *Capanna et al. (2012)* reported a positive and strong correlation between inhibition and cooperativeness. Besides these studies, the literature is quite sparse. There are however, theoretical grounds to

hypothesize that there is a correlation between EC, as defined in Rothbart's model (*Rothbart & Bates, 2006*; *Rothbart et al., 2003*), and cooperativeness. To be characterized as socially tolerant or helpful, a person needs a good temperamental basis of inhibition and attention shifting. Although there is a lack of empirical research linking cooperativeness and EC, there is some literature linking agreeableness and EC (*Evans & Rothbart, 2007*; *Tangney, Baumeister & Boone, 2004*).

High agreeableness is defined as being trusting (*i.e.*, holding positive generalized beliefs about others), compassionate (*i.e.*, active emotional concern for others' well-being), and respectful (*i.e.*, treating others with regard for their personal preference and rights), while inhibiting antagonistic and aggressive impulses (*Soto & John, 2017*). Agreeableness is empirically correlated with EC (*Duckworth, Tsukayama & Kirby, 2013*; *Evans & Rothbart, 2007*; *Kornienko et al., 2018*; *Shiner, 2015*; *Tangney, Baumeister & Boone, 2004*). Among adults, positive and medium correlations were found (*Evans & Rothbart, 2007*; *Tangney, Baumeister & Boone, 2004*). Despite the findings described above, these connections have not been studied in the context of parenting; therefore, more research is needed.

## Effortful control and self-directedness/conscientiousness

High self-directedness is defined as being responsible, purposeful, resourceful, self-accepting, and disciplined; having a high level of willpower and an internal locus of control; identifying the imaginal self as an integrated, purposeful whole person (*Cloninger & Svrakic, 1997*). Empirically, adult EC has been found to correlate positively with self-directedness (*Capanna et al., 2012*; *Cloninger et al., 2019*; *Evans & Rothbart, 2007*; *Waegeman, Declerck & Boone, 2014*). These findings suggest that being highly self-directed is accompanied by a better ability to initiate behavior to achieve long-term goals (*i.e.*, activation control), stay focused, shift attention when needed (*i.e.*, attentional control), and inhibit impulses or responses (*i.e.*, inhibitory control). It is theoretically justified because being responsible, purposeful, and disciplined are mature traits of temperamental self-regulation. Although using the TCI model in order to investigate this correlation between temperament and character is reasonable, the literature is richer when using conscientiousness as the indicator of character (see review by *Bridgett et al. (2015)*).

High conscientiousness is defined as being organized (*i.e.*, preference for order and structure), productive (*i.e.*, having a work ethic and persistence while pursuing goals) and responsible (*i.e.*, committed to meeting duties and obligations; *Soto & John, 2017*). There is theoretical work showing that EC is the temperamental foundation of conscientiousness (see *Eisenberg et al. (2014)*, review by *Bridgett et al. (2015)*; chapter by *Jackson & Hill (2019)*). This notion has also been supported empirically (*i.e.*, *Atherton, Lawson & Robins, 2020*; *Evans & Rothbart, 2007*; *Jensen-Campbell et al., 2002*). However, most of the literature focused on children and their developmental cascade from EC to Conscientiousness, and the research on adults did not investigate parents.

All four character traits presented above are related to EC and there is a theoretical basis to presume that EC is their basis to some extent. As people grow and become parents, their personality (which is the combination of their temperament and character) is the foundation for the development of their parenting practices.

## Cooperativeness, agreeableness and parenting practices

Parents with (self-reported) high levels of Cooperativeness are more inclined to exert authoritative and caring parenting (*Kitamura et al., 2009*; *Zohar, Lev Ari & Bachner-Melman, 2018*). The empirical findings linking agreeableness to parental practice are even better established. Agreeableness is correlated positively with adaptive practices such as warmth, autonomy support (*de Haan, Deković & Prinzie, 2012*; see meta-analysis by *Prinzie et al. (2009)*), and with more adaptive parenting style (*Bahrami et al., 2018*; *Sahithya & Raman, 2021*), and negatively correlated with maladaptive practices such as erratic, harsh, and over-reactive (*e.g.*, screaming or yelling) practices (*de Haan, Prinzie & Deković, 2009*; *de Haan, Deković & Prinzie, 2012*).

Focusing on cooperativeness, there is literature linking parental care and child's cooperativeness (*Baker & Verrocchio, 2013*; *Davidov et al., 2022*; *Josefsson et al., 2013*; *Takeuchi et al., 2011*) or parental cooperativeness with children's behavior (*Choi, Hatton-Bowers & Shin, 2022*; *Vachapurath & Ponnuswamy, 2022*). However, the association between parental cooperativeness and parenting is yet to be established. It is quite reasonable to hypothesize that the parent's cooperativeness is not directly connected to their children's behavior, rather, it is mediated through their parenting practices.

## Self-directedness/conscientiousness and parenting practices

High self-directedness was also found to be associated with more authoritative and caring parenting (*Kitamura et al., 2009*; *Zohar, Lev Ari & Bachner-Melman, 2018*). Similar findings were found regarding conscientiousness; it was found to be related to maternal warmth (*Zhang et al., 2019*) and to more authoritative parenting style (*Bahrami et al., 2018*; *Sahithya & Raman, 2021*).

To summarize, the extant literature suggests that there are positive correlations between each pair of the following three: EC and adaptive parenting practices (and negative correlations with maladaptive parenting), EC and character, and character and parenting practices. However, a comprehensive model including all of the three variables, and specifically, the possible mediation role of character, hasn't been tested. Therefore, one of the aims of the present study was to explore the relations between EC, character, and adaptive and maladaptive parenting practices, with the hypothesis being that character would mediate the connection between self-regulation and parenting practices. This comprehensive model is important in order to investigate the paths that lead to children's behavior, which is a complex construct and could be manifested as conduct problems.

## Children's conduct problems

Conduct problems are composed of a broad range of "acting-out" behaviors (*McMahon, Wells & Kotler, 2006*). Conduct problems have high empirical loadings of disobedience and temper, and tendencies to fight, steal and lie (*Goodman, 1997*; *Muris, Meesters & van den Berg, 2003*); hence, these behaviors function as the theoretical and empirical definition of children's conduct problems in the current study. Conduct problems during childhood correlate with criminal behavior and substance abuse in adulthood (*Bevilacqua et al., 2018*; *Erskine et al., 2016*; *Fergusson & Horwood, 1998*; *Kratzer & Hodgins, 1997*), as well as of

educational under-attainment and unemployment (*Fergusson & Horwood, 1998*). Studies and theories about the ontology of conduct problems support the importance of biological predisposition (like EC) and parenting practices (see *Dodge & Pettit (2003)* for the comprehensive model; *Karreman et al., 2009*; *Ruchkin et al., 2001*; *Shaw et al., 2003*; *Shaw & Shelleby, 2014*). Also supporting these notions are empirical findings regarding a mediation effect between parent EC and child conduct problems *via* parenting practices (*Valiente, Lemery-Chalfant & Reiser, 2007*). However, it is not yet known if parents' mature character traits also mediate these associations.

### Study hypotheses

The current study aims to empirically investigate the possible effects of mothers' character traits on parenting practices and children's conduct problems. Thus, the following hypotheses will be tested: (1) Mothers' character traits will mediate the relationship between mothers' EC and mothers' parenting practices. (2) Mothers' EC will mediate children's conduct problems possibly through character traits and parenting practices (see Fig. 1).

## MATERIALS AND METHODS

### Participants

A representative Israeli sample of 387 Jewish mothers of neurotypical kindergarten-aged children was recruited *via* an external recruiting company (iPanel). This company provides online data collection services; it provides a representative sample of the population, and is highly regarded and extensively used for academic research.

The mothers' ages ranged between 23–61 years (M = 37.00 ± 5.02); years of education ranged between 10–28 years (M = 16.10 ± 2.72). Most of the mothers were the biological mothers (97.4%; $n = 377$) and their household economic situation was average (for detailed socio-demographic status, see Table 1). Their children's sex was evenly distributed (51% males; $n = 199$) and their mean age was 5.58 years ± 0.42 (4.49–6.81). Hence, the sample was socioeconomically and geographically diverse.

### Ethical standards and procedure

The study was approved by the Human Subjects Research Committee of Ben-Gurion University of the Negev, protocol number 2211-1, and supported by the Israel Scientific Foundation (ISF; Grant number 533/20). Ethical standards were followed throughout the study. Participants signed an online informed consent and only the recruiting company knew their identity; hence, for the researchers, all the participants were anonymous and privacy was ensured. Upon completing their participation, participants received an agreed payment by the recruiting company.

### Measures

Participants answered a survey built on the Qualtrics platform that included: (1) the adult temperament questionnaire (ATQ–short form; *Evans & Rothbart, 2007*), (2) the coping with children negative emotions scale (CCNES; *Fabes, Eisenberg & Bernzweig, 1990*),

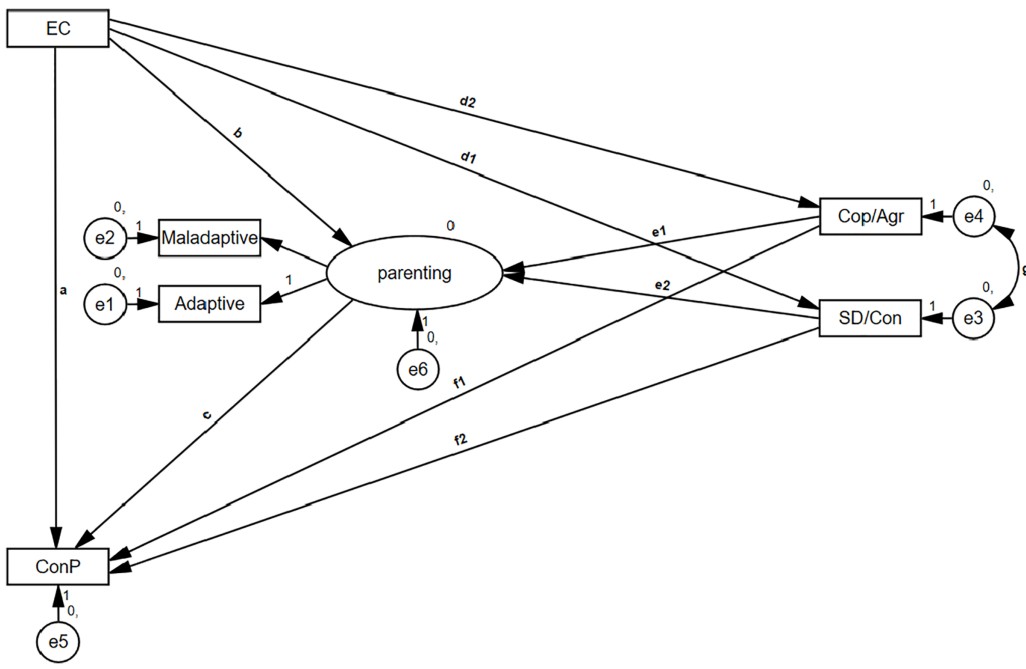

**Figure 1 Hypothesized model.** Conceptual mediations model. EC, effortful control (mother's); maladaptive and adaptive refers to parenting practices; Cop/Agr, cooperativeness or agreeableness (mother's); SD/Con, self-directedness or conscientiousness (mother's); ConP, conduct problems (child's).

**Table 1 Socio-demographic descriptive statistics of the sample.**

| Variable | Frequencies | Variable | Md (Mo) | Range |
|---|---|---|---|---|
| Household income | | Family size | 5 (4) | 1–12 |
| Extremely below average | 63 (16%) | Num of rooms | 4 (4) | 2–9 |
| Below average | 95 (25%) | Num of cars | 1 (1) | 0–4 |
| Average | 115 (30%) | | | |
| Above average | 91 (24%) | | | |
| Extremely above average | 22 (6%) | | | |
| Family status | | | | |
| Single | 20 (5%) | | | |
| Married | 339 (88%) | | | |
| Divorced | 15 (4%) | | | |
| Other | 13 (3%) | | | |

(3) the agreeableness and conscientiousness sub-scales from the big-five inventory (BFI; *John, Donahue & Kentle, 1991*), (4) the cooperativeness and self-directedness sub-scales from the temperament and character inventory-revised (TCI-R; *Cloninger & Svrakic, 1997*), (5) the children behavior questionnaire (CBQ; *Putnam & Rothbart, 2006*), (6) the strength and deficits questionnaire (SDQ; *Goodman, 1997*), and (7) the confusion, hubbub, and order scale (CHAOS; *Matheny et al., 1995*). Afterwards they participated in a short

**Table 2 Psychometric properties of the study variables.**

| Variables | M | SD | α | Range Potential | Actual | Skewness | Kurtosis |
|---|---|---|---|---|---|---|---|
| Mothers' personality | | | | | | | |
| Effortful control | 4.06 | 0.66 | 0.74 | 1–6 | 1.88–6.00 | −0.27 | 0.07 |
| Agreeableness | 3.85 | 0.61 | 0.80 | 1–5 | 1.89–5.00 | −0.59 | 0.10 |
| Conscientiousness | 3.96 | 0.57 | 0.72 | 1–5 | 2.33–5.00 | −0.26 | −0.48 |
| Cooperativeness | 3.84 | 0.46 | 0.82 | 1–5 | 1.95–4.85 | −0.48 | 0.35 |
| Self-directedness | 3.62 | 0.54 | 0.86 | 1–5 | 1.65–4.80 | −0.43 | 0.27 |
| Mothers' parenting practices | | | | | | | |
| Problem-focused responses | 5.47 | 0.94 | 0.87 | 1–7 | 2.25–7.00 | −0.67 | 0.36 |
| Emotion-focused responses | 5.47 | 0.95 | 0.88 | 1–7 | 2.00–7.00 | −0.62 | 0.27 |
| Expressive encouragement responses | 4.74 | 1.22 | 0.92 | 1–7 | 1.25–7.00 | −0.39 | −0.30 |
| Minimization responses | 2.69 | 0.93 | 0.83 | 1–7 | 1.00–6.17 | 0.73 | 0.61 |
| Punitive responses | 2.05 | 0.87 | 0.86 | 1–7 | 1.00–5.42 | 1.15 | 1.01 |
| Distress responses | 2.94 | 0.81 | 0.75 | 1–7 | 1.08–5.67 | 0.30 | 0.22 |
| Childrens' behavior | | | | | | | |
| Conduct problems | 0.32 | 0.35 | 0.66 | 0–2 | 0.00–1.80 | 1.19 | 1.10 |

online behavioral task that combined Go/NoGo and Stroop-like paradigms. The hypotheses of the present article were tested based on the ATQ, CCNES, BFI, TCI and SDQ questionnaires, as described in the following sections. The authors have permission to use these instruments from the copyright holders.

### Mothers' effortful control

Assessment was done using a sub-scale from the adult temperament questionnaire (ATQ–short form; *Evans & Rothbart, 2007*), which is the mean value of 18 items (Cronbach's α = 0.74; see Table 2 for psychometric properties).

### Mothers' character

**Agreeableness and conscientiousness.** Assessment was done using sub-scales from the big-five inventory (BFI; *John, Donahue & Kentle, 1991*). A mean value of nine items per sub-scale was computed. For agreeableness (Agr), Cronbach's α = 0.80, and for conscientiousness (Con), Cronbach's α = 0.72 (see Table 2 for psychometric properties).
**Cooperativeness and self-directedness.** Assessment was done using sub-scales from the temperament and character inventory-revised (TCI-R; *Cloninger & Svrakic, 1997*; *Zohar & Cloninger, 2011*). A mean value of 20 items per sub-scale was computed. For cooperativeness (CO), Cronbach's α = 0.82, and for self-directedness (SD), Cronbach's α = 0.86 (see Table 2 for psychometric properties).

### Parental practice

In order to measure parental practice, the coping with children's negative emotions scale (CCNES; *Fabes, Eisenberg & Bernzweig, 1990*) was used (72 statements). It was used in the

**Table 3 Pearson's correlation matrix between the study variables.**

|  | 1 | 2 | 3 | 4 | 5 | 6 | 7 | 8 | 9 | 10 | 11 |
|---|---|---|---|---|---|---|---|---|---|---|---|
| 1. EC | – | | | | | | | | | | |
| 2. Agr | −0.30*** | – | | | | | | | | | |
| 3. Con | 0.60*** | 0.45*** | – | | | | | | | | |
| 4. CO | 0.36*** | 0.69*** | 0.42*** | – | | | | | | | |
| 5. SD | 0.47*** | 0.48*** | 0.58*** | 0.63*** | – | | | | | | |
| 6. PF | 0.21*** | 0.23*** | 0.23*** | 0.31*** | 0.21*** | – | | | | | |
| 7. EF | 0.15** | 0.24*** | 0.20** | 0.25*** | 0.16*** | 0.84*** | – | | | | |
| 8. EE | 0.21*** | 0.24*** | 0.19** | 0.26*** | 0.22*** | 0.66*** | 0.53*** | – | | | |
| 9. MR | −0.21*** | −0.27*** | −0.18** | −0.34*** | −0.29*** | −0.06 | 0.06 | −0.25*** | – | | |
| 10. PR | −0.22** | −0.29*** | −0.25*** | −0.41*** | −0.36*** | −0.27*** | −0.17*** | −0.35*** | 0.78*** | – | |
| 11. DR | −0.33*** | −0.40*** | −0.30*** | −0.41*** | −0.49*** | −0.35*** | −0.28*** | −0.37*** | 0.47*** | 0.59*** | – |
| 12. ConP | −0.21*** | −0.27*** | −0.26*** | −0.30*** | −0.23*** | −0.16*** | −0.19*** | −0.07 | 0.13* | 0.24*** | 0.18 |
| 13. AP | 0.22*** | 0.27*** | 0.23*** | 0.31*** | 0.22*** | AP and MP $r_p$ = −0.30*** | | | | | |
| 14. MP | −0.29*** | −0.37*** | −0.28*** | −0.44*** | −0.43*** | | | | | | |

Notes:

$N = 387$ ($N = 249$ for correlations with big-five variables). For all scales, higher scores are indicative of more extreme responding in the direction of the construct assessed. EC, mother's effortful control; Agr, mother's agreeableness; Con, mother's conscientiousness; CO, mother's cooperativeness; SD, mother's self-directedness; PF, problem-focused response; EF, emotion-focused response; EE, expressive encouragement response; MR, minimization response; PR, punitive response; DR, distress response; ConP, child's conduct problems; AP, adaptive practices; MP, maladaptive practices.
* $p < 0.05$.
** $p < 0.01$.
*** $p < 0.001$, two-tailed.

current study to measure parents' adaptive and maladaptive practices in different scenarios involving their kindergarten-age child. Computed adaptive sub-scales were problem-focused responses (PF; helping the child solve the problem that caused the distress, $\alpha = 0.87$), emotion-focused responses (EF; helping the child feel better, $\alpha = 0.88$), and expressive encouragement responses (EE; actively encouraging children's expression of negative emotions, $\alpha = 0.92$). Computed maladaptive sub-scales were minimization responses (MR; discounting or devaluing the child's negative emotions/problem, $\alpha = 0.83$), punitive responses (PR; using verbal or physical punishment to control the expression of negative emotion, $\alpha = 0.86$), and distress responses (DR; becoming adversely aroused/distressed by a child's negative emotion, $\alpha = 0.75$). Observed variables were the mean value of each sub-scale, where higher values indicate higher levels of reported constructs (see Table 2 for psychometric properties). In light of high correlations between the adaptive and maladaptive practices (see Table 3; $0.53 < r_p < 0.84$, for adaptive practices correlations; $0.47 < r_p < 0.78$, for maladaptive practices correlations) and similar descriptive statistics (see Table 2), two general mean values were computed and used: (1) adaptive practices composed from PF, EF and EE, and (2) maladaptive practices composed from MR, PR, and DR.

### Children's conduct problems

Assessment was done using a sub-scale from the Strengths and Difficulties Questionnaire (SDQ; *Goodman, 1997*), which is the mean value of five items (Cronbach's $\alpha = 0.66$; see

Table 2 for psychometric properties) regarding disobedience, temper, tendencies to fight, steal, and lie.

## Analytic plan

The data was collected online *via* Qualtrics (2020) and statistical analysis was done using SPSS 27 and AMOS 27 software. Participants who did not complete the questionnaires or failed the alertness checks were removed from the sample. Figure 1 presents the empirical mediation model that was designed to test the study hypothesis. Four observed variables and one latent variable were used for parental practices, using reported adaptive and maladaptive practices. A four-step hierarchical SEM model was constructed using maximum likelihood estimates. The first model constrained all paths, except for the direct path between mothers' EC and children's conduct problems (path a). In the second model, the paths through parental practices were freed (paths b and c). It enabled testing the indirect effect between mothers' EC and children's conduct problems through parenting. In the third model, the paths through the character traits to the parental practice were freed (paths d1, d2, e1, e2, g). It enabled testing the indirect effect between mother's EC and parental practices through character and the mediated mediation between mother's EC and children's conduct problems through character and parenting practices. Moreover, it allowed assessing the correlation between one's character traits, in accordance with the theoretical and empirical literature. The fourth model was the full model that is depicted in Fig. 1. In order to test for replicability, the same model was tested twice—once with variables from the BFI and once with variables from the TCI.

## Transparency and openness

Sample size was determined using the A-priori sample size calculator for structural equation models at the free statistics calculator (Soper, 2022; https://www.danielsoper.com/statcalc). The recommended minimum sample size for small anticipated effect size (0.2), power of 0.8, alpha of 0.5, one latent variable and six observed variables was 200. All collected data were included in the analysis. In light of the required sample size, we collected the data in two waves, using the services of the same company both times in order to prevent double participation by the same participant. All data and the analysis code are available (https://zenodo.org/badge/latestdoi/554084676). The study design and analysis were not pre-registered.

## RESULTS

Table 3 presents Pearson correlations between the study variables as expected, the correlations are significant and in the expected directions. Moreover, correlations between background variables and the study variables were tested. As expected, most correlations were not significant, besides the correlations between the sex of the child, number of household members and adaptive parenting practices ($N = 387$, $r_{bs} = -0.14$, $p = 0.004$; $N = 387$, $r_p = -0.10$, $p = 0.044$) and the correlation between number of education years and maladaptive parenting practices ($N = 387$, $r_p = -0.12$, $p = 0.019$). Hence, these background variables were controlled for in the SEM analyses, when predicting the factor of parenting

**Table 4 Four-step hierarchical structural equation modeling of mediation effects between mothers' effortful control and children's conduct problems.**

| Path | Model 1 | | | Model 2 | | | Model 3 | | | Model 4 | | |
|---|---|---|---|---|---|---|---|---|---|---|---|---|
| | *B* | *(SE)* | β | *B* | *(SE)* | β | *B* | *(SE)* | β | *B* | *(SE)* | β |
| EC and ConP (a) | −0.11 | (0.03) | −0.21*** | 0.04 | (0.04) | −0.08 | −0.03 | (0.03) | −0.05 | −0.05 | (0.03) | −0.09 |
| EC and parenting (b) | 0 | | | 0.29 | (0.06) | 0.46*** | 0.10 | (0.04) | 0.17** | 0.10 | (0.04) | 0.16** |
| Parenting and ConP (c) | 0 | | | −0.23 | (0.08) | −0.29** | −0.30 | (0.07) | −0.35*** | −0.16 | (0.11) | −0.18 |
| EC and SD (d1) | 0 | | | 0 | | | 00.39 | (0.04) | 0.47*** | 0.39 | (0.04) | 0.47*** |
| EC and Cop (d2) | 0 | | | 0 | | | 0.25 | (0.03) | 0.36*** | 0.25 | (0.03) | 0.36*** |
| Cop and parenting (e1) | 0 | | | 0 | | | 0.44 | (0.08) | 0.50*** | 0.39 | (0.08) | 0.44*** |
| SD and parenting (e2) | 0 | | | 0 | | | 0.17 | (0.06) | 0.23** | 0.19 | (0.07) | 0.26** |
| Cop and ConP (f1) | 0 | | | 0 | | | 0 | | | −0.13 | (0.06) | −0.17* |
| SD and ConP (f2) | 0 | | | 0 | | | 0 | | | 0.02 | (0.05) | 0.03 |
| Covariance | | | | | | | | | | | | |
| Cop and SD (g) | 0 | | | 0 | | | 0.11 | (0.01) | 0.56*** | 0.11 | (0.01) | 0.56*** |
| Error terms | | | | | | | | | | | | |
| e1 | 0.13 | | | 0.65 | | | 0.68 | | | 0.67 | | |
| e2 | 0.50 | | | 0.33 | | | 0.33 | | | 0.30 | | |
| e3 | 0.30 | | | 0.30 | | | 0.23 | | | 0.23 | | |
| e4 | 0.21 | | | 0.21 | | | 0.18 | | | 0.18 | | |
| e5 | 0.11 | | | 0.11 | | | 0.10 | | | 0.11 | | |
| e6 | 0.71 | | | 0.15 | | | 0.07 | | | 0.08 | | |
| Model fit | | | | | | | | | | | | |
| NFI | 0.109 | | | 0.215 | | | 0.985 | | | 0.991 | | |
| CFI | 0.108 | | | 0.211 | | | 0.994 | | | 0.997 | | |
| TLI | −0.029 | | | −0.183 | | | 0.983 | | | 0.986 | | |
| RMSEA | 0.295 | | | 0.317 | | | 0.038 | | | 0.035 | | |
| AIC | 480.37 | | | 430.79 | | | 51.74 | | | 52.38 | | |

**Notes:**
*N* = 387. For all scales, higher scores are indicative of more extreme responding in the direction of the construct assessed. Letters in parenthesis correspond with path letters in Fig. 1. EC, effortful control; maladaptive and adaptive parenting practices; Cop, cooperativeness; SD, self-directedness; ConP, conduct problems. Scaler was adaptive practices for mother's parenting.
\* $p < 0.05$.
\*\* $p < 0.01$.
\*\*\* $p < 0.001$.

practices. It was found that controlling for the background did not change the pattern of results and that the background variables were found to be insignificant (see Supplemental Analyses). In light of these results, the analyses presented here are without the background variables.

In order to test the study hypothesis, a four-step hierarchical SEM model was constructed using maximum likelihood estimates. To start with, character was measured using cooperativeness and self-directedness. Table 4 presents evidence supporting that the third model is the most fitting model, with CFI = 0.994; NFI = 0.985; TLI = 0.983; RMSEA = 0.038; $\Delta AIC_{Model\,2-Model\,3}$ = 379.05. In the first model, the direct path between mothers' EC and their child's conduct problems was found to be negative and significant

($\beta_a = -0.21$; $p < 0.001$). In the third model, this direct path was found to be insignificant ($\beta_{a'} = -0.03$; $p = 0.398$). In order to test the significance of mediations, the bias-corrected bootstrap confidence interval (BCBCI) was calculated for seven mediation effects:

1. $Bparenting = B_b * B_c$
2. $Bcooperativness = B_{d2} * B_{e1}$
3. $Bself\ directedness = B_{d1} * B_{e2}$
4. $Bcharacter = B_{d2} * B_{e1} + B_{d1} * B_{e2}$
5. $Bcoperativenss + parenting = B_{d2} * B_{e1} * B_c$
6. $Bself\ directedness + parenting = B_{d1} * B_{e2} * B_c$
7. $Bcharacter + parenting = B_{d2} * B_{e1} * B_c + B_{d1} * B_{e2} * B_c$

See summarized results in Table 5; specifically, the indirect path through the parenting practices was found to be significant ($B = -0.03$, 95% BCBCI [$-0.08$ to $-0.01$]; $p = 0.017$), as well as the path between EC and parenting practices through Cooperativeness ($B = 0.11$, 95% BCBCI [0.06–0.18]; $p < 0.001$) and self-directedness ($B = 0.07$, 95% BCBCI [0.02–0.12]; $p = 0.011$). Moreover, the mediation between mothers' EC and children's conduct problems through cooperativeness and parenting was found to be significant ($B = -0.03$, 95% BCBCI [$-0.06$ to $-0.01$]; $p < 0.001$), as well as mediation through self-directedness and parenting ($B = -0.02$, 95% BCBCI [$-0.04$ to $-0.01$]; $p = 0.010$). These findings suggest a full mediation (Fig. 2 presents factor loadings and path's beta).

In order to validate study findings, the same analysis was repeated using the big-five indices as indicators of character. Table 6 presents evidence supporting that the third model is the most fitting model, with CFI = 0.996; NFI = 0.978; TLI = 0.988; RMSEA = 0.029; $\Delta AIC_{Model\,2 - Model\,3} = 195.71$. In the first model, the direct path between mothers' EC and their child's conduct problems was found to be negative and significant ($\beta_a = -0.17$; $p = 0.005$). In the third model, this direct path was found to be insignificant ($\beta_{a'} = -0.02$; $p = 0.836$). In order to test mediation significance, the bias-corrected bootstrap confidence interval (BCBCI) was calculated for seven mediation effects. See summarized results in Table 5; specifically, the indirect path through the parenting practices was found to be marginally significant ($B = -0.04$, 95% BCBCI [$-0.12$, 0.00]; $p = 0.058$), as well as the path between EC and parenting practices through agreeableness ($B = 0.10$, 95% BCBCI [0.05, 0.17]; $p < 0.001$). However, the path through conscientiousness was found to be insignificant ($B = 0.07$, 95% BCBCI [$-0.03$ to 0.17]; $p = 0.137$). Moreover, the mediation between mothers' EC and children's conduct problems through agreeableness and parenting was found to be significant ($B = -0.03$, 95% BCBCI [$-0.05$ to $-0.01$]; $p = 0.003$). However, the path through conscientiousness and parenting was found to be insignificant ($B = -0.02$, 95% BCBCI [$-0.07$, 0.01]; $p = 0.151$). These findings suggest a full mediation through agreeableness and parenting only (Fig. 3 presents factor loadings and path's beta).

**Table 5 Mediation estimates and significance tests.**

| Mediator | TCI model | | | | BFI model | | | |
|---|---|---|---|---|---|---|---|---|
| | B | 95% BCBCI | | p value | B | 95% BCBCI | | p value |
| | | LL | UL | | | LL | UL | |
| Parenting (bc) | −0.03 | −0.08 | −0.01 | 0.017 | −0.04 | −0.12 | 0.00 | 0.058 |
| Cooperativeness/Agreeableness (d2e1) | 0.11 | 0.06 | 0.18 | <0.001 | 0.10 | 0.05 | 0.17 | <0.001 |
| Self-directedness/Conscientiousness (d1e2) | 0.07 | 0.02 | 0.12 | 0.011 | 0.07 | −0.03 | 0.17 | 0.137 |
| Character (d2e1 + d1e2) | 0.18 | 0.11 | 0.25 | <0.001 | 0.17 | 0.07 | 0.28 | 0.001 |
| Cooperativeness/Agreeableness and parenting (d2e1c) | −0.03 | −0.06 | −0.01 | <0.001 | −0.03 | −0.05 | −0.01 | 0.003 |
| Self-directedness/Conscientiousness and parenting (d1e2c) | −0.02 | −0.04 | −0.01 | 0.010 | −0.02 | −0.07 | 0.01 | 0.151 |
| Character and parenting (d2e1c + d1e2c) | −0.05 | −0.09 | −0.03 | <0.001 | −0.05 | −0.10 | −0.01 | 0.005 |

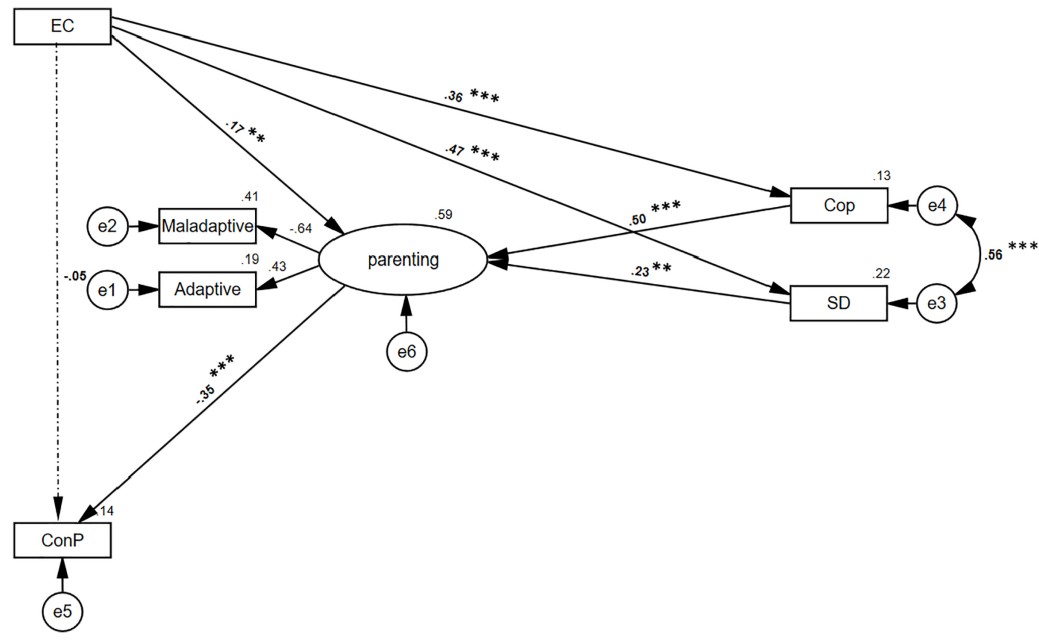

**Figure 2 Structural equations modeling of mediations hypothesis using the TCI indexes.** $N = 387$. For all scales, higher scores are indicative of more extreme responding in the direction of the construct assessed. Path value is beta. Model fit: CFI = 0.994; NFI = 0.985; TLI = 0.983; RMSEA = 0.038. EC, effortful control; maladaptive and adaptive refers to parenting practices; Cop, cooperativeness; SD, self-directedness; ConP, conduct problems. Scaler was adaptive practices for mother's parenting. ***$p < 0.001$, **$p < 0.01$.

## DISCUSSION

This present study is the first to empirically test the possible mediation effect of mothers' character traits and parenting practices, on the connection between maternal EC and child conduct problems. The hypotheses were tested in a representative sample of Jewish Israeli mothers of kindergarten-aged children. The study findings suggest a partial mediation *via* mothers' character traits of the relationship between mothers' EC and parenting practices.

**Table 6 Four-step hierarchical structural equation modeling of mediation effects between mothers' effortful control and children's conduct problems.**

| Path | Model 1 | | | Model 2 | | | Model 3 | | | Model 4 | | |
|---|---|---|---|---|---|---|---|---|---|---|---|---|
| | **B** | **(SE)** | **β** | **B** | **(SE)** | **β** | **B** | **(SE)** | **β** | **B** | **(SE)** | **β** |
| EC and ConP (a) | −0.09 | (0.03) | −0.17** | −0.04 | (0.04) | −0.08 | −0.01 | (0.04) | −0.02 | 0.01 | (0.04) | 0.01 |
| EC and parenting (b) | 0 | | | 0.30 | (0.08) | 0.41*** | 0.13 | (0.07) | 0.19[+] | 0.15 | (0.07) | 0.21* |
| Parenting and ConP (c) | 0 | | | −0.18 | (0.08) | −0.26** | −0.28 | (0.08) | −0.36*** | −0.11 | (0.09) | −0.15 |
| EC and Con (d1) | 0 | | | 0 | | | 0.52 | (0.04) | 0.60*** | 0.52 | (0.04) | 0.60*** |
| EC and Agr (d2) | 0 | | | 0 | | | 0.27 | (0.06) | 0.30*** | 0.27 | (0.06) | 0.30*** |
| Agr and parenting (e1) | 0 | | | 0 | | | 0.35 | (0.08) | 0.47*** | 0.33 | (0.08) | 0.43*** |
| Con and parenting (e2) | 0 | | | 0 | | | 0.14 | (0.08) | 0.17 | 0.09 | (0.09) | 0.11 |
| Agr and ConP (f1) | 0 | | | 0 | | | 0 | | | −0.07 | (0.05) | −0.13 |
| Con and ConP (f2) | 0 | | | 0 | | | 0 | | | 0.01 | (0.05) | −0.15[+] |
| Covariance | | | | | | | | | | | | |
| Agr and Con (g) | 0 | | | 0 | | | 0.09 | (0.02) | 0.35*** | 0.09 | (0.02) | 0.35*** |
| Error terms | | | | | | | | | | | | |
| e1 | 0.10 | | | 0.60 | | | 0.63 | | | 0.61 | | |
| e2 | 0.51 | | | 0.34 | | | 0.37 | | | 0.33 | | |
| e3 | 0.32 | | | 0.32 | | | 0.21 | | | 0.21 | | |
| e4 | 0.37 | | | 0.37 | | | 0.33 | | | 0.33 | | |
| e5 | 0.11 | | | 0.11 | | | 0.10 | | | 0.10 | | |
| e6 | 0.73 | | | 0.20 | | | 0.11 | | | 0.14 | | |
| Model fit | | | | | | | | | | | | |
| NFI | 0.136 | | | 0.240 | | | 0.978 | | | 0.998 | | |
| CFI | 0.136 | | | 0.235 | | | 0.996 | | | 1 | | |
| TLI | 0.004 | | | −0.147 | | | 0.988 | | | 1 | | |
| RMSEA | 0.266 | | | 0.285 | | | 0.029 | | | 0 | | |
| AIC | 270.77 | | | 245.79 | | | 50.08 | | | 48.60 | | |

**Notes:**
$N = 249$. For all scales, higher scores are indicative of more extreme responding in the direction of the construct assessed. Letters in parenthesis corresponds with paths letters in Fig. 1. EC, effortful control; maladaptive and adaptive parenting practices; Agr, agreeableness; Con, conscientiousness; ConP, conduct problems. Scaler was Adaptive practices for mother's parenting.
[+] $p < 0.10$.
[*] $p < 0.05$.
[**] $p < 0.01$.
[***] $p < 0.001$.

Moreover, it suggests a full mediation *via* character traits and parenting practices of the relationship between mothers' EC and children's conduct problems. Empirical differences were found when character was measured using concepts from the TCI or the BFI. These models imply that the temperamental component of EC is the basis for the development of mature characteristics of personality. These traits are then manifested in mothers' parental practices, which relate to children's behavior. Finally, the study findings show that while self-directedness acts as a mediator, conscientiousness does not.

The empirical findings presented here elaborate *Bridgett et al. (2015)*'s model, which suggested that the transgenerational transmission of self-regulation between parents and their children passes through their parenting practices, among other elements. The current

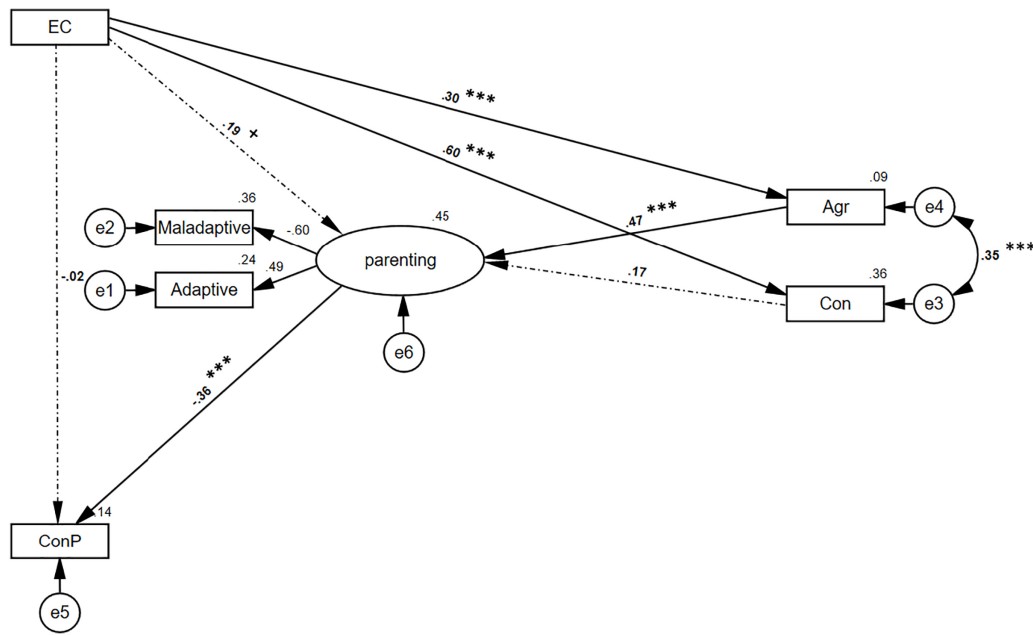

**Figure 3 Structural equations modeling of mediations hypothesis using the BFI indexes.** *N* = 249. For all scales, higher scores are indicative of more extreme responding in the direction of the construct assessed. Path value is beta. Model fit: CFI = 0.996; NFI = 0.978; TLI = 0.988; RMSEA = 0.029. EC, effortful control; maladaptive and adaptive refers to parenting practices; Agr, agreeableness; Con, conscientiousness; ConP, conduct problems. Scaler was adaptive practices for mother's parenting. ***$p < 0.001$, +$p < 0.1$.

findings support and expand the theoretical relationship between these factors, as empirical findings support the relationship between higher EC and more adaptive practices (*Bridgett et al., 2011*, *2015*; *Distefano et al., 2018*; *Rutherford et al., 2015*; *Shaffer & Obradović, 2017*) and less maladaptive ones (*Bridgett et al., 2013*, *2017*; *Crandall et al., 2016*; *Davenport et al., 2011*). Our findings indicate that such a relationship between mothers' EC and their parental practice is actually indirect, and it is expressed through the mother's mature character, as has been suggested in the literature regarding conscientiousness (*Atherton, Lawson & Robins, 2020*; *Bridgett et al., 2015*; *Eisenberg et al., 2014*; *Evans & Rothbart, 2007*; *Jackson & Hill, 2019*; *Jensen-Campbell et al., 2002*). The current study expands these findings and suggests that the temperamental tendency to utilize attentional resources and to inhibit behavioral responses for regulating emotions and related behaviors (*Rothbart, Derryberry & Posner, 1994*) is the foundation for more mature constructs of character, such as being more self-directed (*i.e.*, responsible, purposeful, self-accepting, and disciplined; *Cloninger & Svrakic, 1997*).

The study findings also elaborate on the existing literature and suggest that temperamental EC underlies not only conscientiousness or self-directedness, but also socially oriented traits, such as agreeableness and cooperativeness. The fundamental ability of EC can be expressed in being more cooperative (*i.e.*, socially tolerant, empathic, helpful, and compassionate; *Cloninger & Svrakic, 1997*) and agreeable (*i.e.*, trusting, compassionate and respectful, while inhibiting antagonistic and aggressive impulses; *Soto & John, 2017*).

These positive relationships between EC and all of the character traits do support the existence of a more resilient type of personality configuration as suggested previously (*Cloninger et al., 2019*; *Cloninger & Zohar, 2011*).

These personality traits form a firm basis for more adaptive parental practices. More specifically, having better abilities of inhibition and attention (better EC) translate into higher levels of agreeableness, cooperativeness, self-directedness, and conscientiousness, which in turn enable the mothers to be more helpful to their children's needs, be more soothing and actively encourage children's expression of negative emotions. Moreover, it helps them to use less of the maladaptive ones, such as minimizing or discounting their child's feelings, as well as less use of verbal or physical punishment to control the expression of negative emotion (*Fabes, Eisenberg & Bernzweig, 1990*).

When addressing the relationship between mothers' EC and children's conduct problems, the current study findings correspond with *Dodge & Pettit's (2003)* and *Bronfenbrenner's (1979)* theories, which suggest that parenting, a contextual feature of the child's environment, might support the development of their conduct problems. The current findings suggest that a mother's temperamental ability of exerting EC is not directly responsible for her children's conduct problems, but elaborate on the research of *Valiente, Lemery-Chalfant & Reiser (2007)* who imply that the mediating factors are parenting practices and children's EC, highlighting the role of parenting in child development. The current findings suggest that a mother's temperamental ability of exerting EC is not directly responsible for her children's conduct problems. The current findings partially replicate their findings and support the notion of an indirect relationship, that is, that mothers' mature character is another important mediator that should be taken into consideration when trying to better understand children's conduct problems. Mothers' mature personality, which is composed of temperamental tendencies to successfully inhibit responses, including high levels of attentional focusing (*i.e.*, high effortful control; *Rothbart et al., 2003*) and her mature characteristics of being more socially tolerant, helpful, purposeful and disciplined (*i.e.*, high agreeableness/ cooperativeness and high self-directedness; *Cloninger & Svrakic, 1997*; *Soto & John, 2017*), is an important predictor of her de-facto parenting that, in turn, corresponds with their children's exhibiting behaviors of disobedience and temper, and tendencies to fight, steal and lie. Thus, the developmental distinction between temperament and character (*Cloninger, 2004*; *Zohar et al., 2019*) is of use in understanding this inter-generational co-evolution of mothers' personality, EC and children' conduct problems.

There were some differences in the mediation of the models when using conscientiousness, as opposed to self-directedness. A higher correlation was found with EC and conscientiousness, than with self-directedness. Secondly, while there was a correlation between self-directedness and parenting practices, but not with conscientiousness. While all three constructs are positively interlinked and confer resilience (*Cloninger et al., 2019*), the differences between the constructs may explain the pattern of intercorrelations (*Capanna et al., 2012*; *Cloninger & Svrakic, 1997*; *Soto & John, 2017*). Conscientiousness relies more heavily on cognitive and attentional constructs as a basis for development, as it is characterized as being more organized, responsible, with high work ethics and pursuing

of goals and is more similar to a temperament trait than to a character trait as conceptualized in the TCI model (*Cloninger, 2004*; *Soto & John, 2017*). Moreover, self-directedness emphasizes the holistic sense of self, that is, identifies the imaginal self as an integrated, purposeful and whole person (*Cloninger & Svrakic, 1997*).

## CONCLUSIONS

This study demonstrated mediation effects of mothers' character traits and parental practices on children's conduct problems. Understanding the variables that explain mothers' parental practices and children's conduct problems has both theoretical and practical importance for the development of tailored-to-individual-differences and effective interventions designed to assist mothers in improving their practices.

This study has several limitations that need to be stated. First, our sample included only mothers, and mainly married ones. More research is needed to generalize the findings to fathers also, in light of empirical findings suggesting that fathers and their children have different relationships from mothers and children (*Gryczkowski, Jordan & Mercer, 2010*; *McKinney & Renk, 2008*; *Rinaldi & Howe, 2012*), and/or additional parenting familial settings. Secondly, the findings are based on reported measures only; while this is standard practice in research on such young children, it is not free of reporter bias. Further research is needed to replicate these results using more objective and behavioral measures. Although the sample had adequate power to test the study hypotheses, and was drawn to be representative, the high correlation between the two mature character traits, self-directedness and cooperativeness, higher than in the normative sample (see *Zohar & Cloninger, 2011*), suggests relatively well-adjusted individuals chose to participate in the current study, and that research would profit from a more diverse sample.

Another direction for further research is elaborating the mediation effect on personality traits such as perfectionism, since perfectionism has been found to predict parenting practices and has important theoretical meaning in the context of parenting (*Azizi & Besharat, 2011*; *Carmo et al., 2021*; *Flett, Hewitt & Singer, 1995*; *Lee, Schoppe-Sullivan & Dush, 2012*). Lastly, while this is a highly plausible model, there are alternate explanations for the relationship between the variables, including passive and evocative gene-environment correlations (*Knafo & Jaffee, 2013*) influencing the association between parents' temperament, parents' character and parenting practices and children's behavioral problems.

### Funding

This work was supported by the Israel Scientific Foundation (ISF; Grant Number 533/20). The funders had no role in study design, data collection and analysis, decision to publish, or preparation of the manuscript.

## Grant Disclosures

The following grant information was disclosed by the authors:
Israel Scientific Foundation: 533/20.

## Competing Interests

Ada H. Zohar is an Academic Editor for PeerJ.

## Author Contributions

- Maor Yeshua conceived and designed the experiments, performed the experiments, analyzed the data, prepared figures and/or tables, authored or reviewed drafts of the article, and approved the final draft.
- Ada H. Zohar conceived and designed the experiments, authored or reviewed drafts of the article, and approved the final draft.
- Andrea Berger conceived and designed the experiments, authored or reviewed drafts of the article, and approved the final draft.

## Data Availability

The raw data and the analyzed models are available at GitHub and Zenodo: https://github.com/MaorYeshua/Mediation-paper.git.

MaorYeshua. (2023). MaorYeshua/Mediation-paper: Mediation effects of personality - Final files (Final_Materials). Zenodo. https://doi.org/10.5281/zenodo.7699615.

## Supplemental Information

Supplemental information for this article can be found online at http://dx.doi.org/10.7717/peerj.15211#supplemental-information.

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
