# Peer review of "Mediating roles of character traits and parenting in the relationship between maternal effortful control and children’s conduct problems"

_PeerJ, doi:10.7717/peerj.15211_

## Round 0.1 · original submission · Minor Revisions

Three reviewers with expertise in psychology, personality assessment, and parenting have provided constructive feedback that will help you in revising with awareness of the reactions of informed readers. Two recommended minor revisions and one major revisions. Overall they all recognized the positive scientific merits but point out ways it can be improved.

A prominent issue is the discussion of the relationship of Effortful Control to personality. There is actually data about the correlations of the ATQ and NEO-PI-R to the TCI-R that will be helpful to you. It was published in Table 4 of Cloninger CR, Cloninger KM, Zwir I, Keltigangas-Jarvinen L. The complex genetics and biology of human temperament: a review of traditional concepts in relation to new molecular findings. Translational Psychiatry, 2019 Nov 11; 9(1):290 (22 pages). https://doi.org/10.1038/s41398-019-0621-4. EC corresponds to high Self-0directedness, high Persistence, and low harm avoidance so it measures resilience. Conscientiousness measures the same profile of features plus low novelty seeking.

A second issue to discuss is the sample. You used a panel recruited by a well-respected external recruiting company, but the correlation between Self-directedness and Cooperativeness seems unusually high compared to the factor structure reported by Zohar in European J of Psychological Assessment 2011 (27:73-80) has cross-factor loading of these two traits of .3 or so rather than a correlation of 0.6. This suggests that the panel provided by the recruiting company may have been selected for being jointly reliable and cooperative, which is not surprising. This is a minor issue given your thoughtful overview of the findings, but it may be worth noting.

·

Basic reporting

No comments. Please see attached document

Experimental design

No comments. Please see attached document

Validity of the findings

No comments. Please see attached document

Additional comments

Please see attached document.

·

Basic reporting

The paper uses clear, unambiguous, professional English throughout and includes professional article structure, figures, and tables. It is a novel stand-alone paper with shared raw data and results related to hypotheses.

Experimental design

Corresponds to the first basic research done within the objectives and scope of the journal.
Research questions and hypotheses are well presented, and preceding studies for this are well reviewed. The study was conducted according to strict ethical standards.
It also presents enough details and explanations to disprove the research

Validity of the findings

All basic data has been provided. The results were presented in accordance with the hypotheses.
The conclusions are well-documented and linked to the original research question. The significance and implications of the study and its academic contributions are also well described.

Additional comments

As the authors emphasize, this study is considered to be the first to examine the mediating effect of character traits and parenting practices between mother's EC and children's conduct problems. In particular, this study is very interesting in that temperamental regulation is the developmental basis of personality represented by Self-Directedness and Cooperativeness, and that it shows an example of how self-regulation between parents and children can be transmitted between generations. Using the concept of EC, it was shown that temperamental regulation affects mothers' parenting practices through character traits, and this parenting in turn affects children's behavior related to temperamental regulation. As known, both temperament and character have genetic contents, but many literatures report that temperament is the foundation of personality development from an interactive point of view. However, it has very special academic value because there are few studies that have empirically explained the role of temperamental regulation with the concept of EC in relation to parenting practices in detail. In particular, the fact that the mother's Self-Directedness and Cooperativeness itself directly affect the child's behavior, but also through parenting behavior, suggests a perspective on improving parental behavior to intervene in the child's conduct problems. I would say it is a very useful resource. Therefore, it is evaluated as a paper with great theoretical and practical contributions.
However, it is suggested that some minor problems such as the following should be supplemented.

1. Abstract: at lines 34-36: It would be more helpful if you describe the research results in more detail (especially the structural equation model verification results) through the main variables used for hypothesis verification.
2. Key words: Write Keywords.
3. In this study, it is not specifically explained which conduct problems of kindergarten children are influenced by the mother's EC (Effortful Control), Character traits, and parenting practices in relation to children's conduct problems, which are one of the main variables. Although the main purpose of this study is to verify the causal relationship between the variables, the actual content dealt with through the variables is also considered important. Please see on this point.
4. Subject of study: It would be helpful to explain why kindergarten-age children were selected while dealing with children's conduct problems. Based on this, future research will be able to study subjects of the same or different ages.
5. Hypothesis verification: Table 2: In addition to skewness, kurtosis is generally presented as a criterion for determining whether variables form a normal distribution. If you add it, it will be more helpful to understand this.
6. Figure 1: 1) Conceptual mediations model: This model is a research model for path analysis. Because EC is an exogenous variable and is not explained by other variables, it does not include errors. Therefore, an error variable should not be set. Deleting it from the figure would help. 2) Correlation between measurement errors of endogenous variables can be set in the process of developing a correction model to increase the model fit, that is, to increase the chi-square value. However, it could be not theoretically correct to assume that there will be correlations between measurement errors when setting up a research model from the beginning.
3) Figure 2, 3: In general, insignificant results are marked with dotted lines. Editing would be helpful for understanding. Please refer to this.

End.

Reviewer 3 ·

Basic reporting

1) I found this paper used clear and unambiguous English throughout the paper. I just make a few minor comments regarding technical usage.
2) The source of line 47 (e.g., Lee et al, 2017) should be provided in the reference list, and make sure other sources are well provided.
3) The use of frequency seems more appropriate instead that of Md and Mo in Table 1.
4) If the significance of values in figures 2 & 3 is provided (e.g., ***), it would be much easier to understand.

Experimental design

no comment.

Validity of the findings

1) Though the experimental design is well-planned and conducted, I think the interpretation of the results is not thoroughly mentioned: that of cooperativeness or agreeableness is not found in the Discussion section. It should be provided. And I suggest authors should offer the meaning of coupled SD + CO for a rich interpretation.

2) Besides, (compared to the introduction), the interpretation of the association between EC and conduct problems should be provided with more previous literature and detailed explanation (e.g., transgenerational transmission of self-regulation).

3) The authors tried to explain the difference between self-directedness and conscientiousness in light of construct or concept and correlation (lines 372-383, lines 362-365) but I think it comes from the more fundamental difference regarding the personality consisting of temperament and character (or TCI vs. BFI). I suggest you refer to the literature of Cloninger and others regarding character development.

Additional comments

no comment

---

## Round 0.2 · accepted · Accept

All reviewers and I agree that this contribution is a substantial contribution to the literature and that you have addressed the major issues raised in the initial review. One reviewer suggested some minor revisions that do not prevent acceptance of the manuscript but you may want to clarify the point about the specific measure of conduct used in kindergarteners (whether "disobedience and temper, and tendencies to fight, steal and lie” were actually measured for kindergarten children). I am satisfied with your presentation of the SEM regarding the correlation between Cooperativeness and Self-directedness, which you represented by correlated residuals. This is acceptable because of the discussion about the sample having an unusually high correlation between these variables due to what appears to have been an over-representation of well-adjusted parents and the fact that any bias in self-reporting is likely to affect both these traits in the same prosocial way.

In summary, the paper is acceptable as it is, but I would encourage you to clarify the point about the measure of conduct problem in finalizing the files during the copy editing process if there was some modification of the standard instrument for this sample. No revision is needed if the standard published instrument was used.

·

Basic reporting

Thank you to the authors for considering all my comments on their manuscript. After reading this second version, it is a pleasure to accept this wonderful manuscript for publication.

Experimental design

No comment

Validity of the findings

No comment

Additional comments

No comment

·

Basic reporting

No comment.
See Additional comments below for Additional supplements.

Experimental design

No comment.
See Additional comments below for Additional supplements.

Validity of the findings

No comment.
See Additional comments below for Additional supplements.

Additional comments

It seems that the authors put a lot of effort into revising the paper. So the paper was generally well revised and supplemented. However, there are a few minor corrections as shown below, so I ask for supplementation and leave it to the authors.

Abstract: Abstracts generally do not include statistical figures when presenting results. I think it would suffice to express the sentence as "The selected path models were found to be good models". Alternatively, since the purpose of this study is to verify the mediating effect, not the model validation itself, you can omit this sentence from the abstract.

About Children's Conduct Problems: The researcher said “We have now clarified this on lines 196-198.” As the answer was given, what the conduct problem defined in this paper is well expressed in this paragraph. However, in reality, it is not expressed whether the conduct problem to be measured in this paper was measured well. In other words, it is necessary to describe in Measures whether “disobedience and temper, and tendencies to fight, steal and lie” were actually measured for kindergarten children. A more specific description of the SDQ instrument in lines 280-283 about the measure of Child Behavior Problems will help the reader understand the behavior problems measured in this study. (e.g. the contents of scales).

Figure correction: Figure 1, Figure 2, and Figure 3: I mentioned that it was wrong to draw the correlation between the error terms in the hypothetical model, but it seems that it was not conveyed. If researchers want to present a “correlation between personality traits according to theoretical and empirical literature” to the model, they should not derive a correlation between the two error terms e4 and e3. If the correlation curves between the error terms are removed and the correlation curves are drawn directly to the observed variables, Cop and SD, and equally to Agr and Con, it meets the researcher's purpose. I hope the same applies to Figure 2 and Figure 3. In fact, the beta value of .56 is the covariance between Cop and SD, as shown in Table 4. In this regard, the following supplementary matters should also be considered.
A complement to hypothesis models: As researchers are well known, every causal and correlation line in a model represents a hypothesis. The correlations between these personality traits should not be plotted in Figure 1 because the researchers did not hypothesize the correlations between the personality traits as observed variables. If the relationship between personality traits is to be set up in the model based on a theoretical basis, the need for this should be presented in the text. I’d suggest that you refer to it.

End.

Reviewer 3 ·

Basic reporting

no comment

Experimental design

no comment

Validity of the findings

no comment

Additional comments

I appreciate that the authors adequately addressed comments and made your paper upgraded. Given that effort, I have no more comments.